# A Novel Force-Sensing Smart Textile: Inserting Silicone-Embedded FBG Sensors into a Knitted Undergarment

**DOI:** 10.3390/s23115145

**Published:** 2023-05-28

**Authors:** Ka-Po Lee, Joanne Yip, Kit-Lun Yick, Chao Lu, Linyue Lu, Qi-Wen Emma Lei

**Affiliations:** 1School of Fashion and Textile, The Hong Kong Polytechnic University, Hong Kong 999077, China; maple.lee@connect.polyu.hk (K.-P.L.); kit-lun.yick@polyu.edu.hk (K.-L.Y.); 19028951d@connect.polyu.hk (Q.-W.E.L.); 2Photonics Research Institute, The Hong Kong Polytechnic University, Hong Kong 999077, China; chao.lu@polyu.edu.hk (C.L.);; 3Department of Electronic and Information Engineering, The Hong Kong Polytechnic University, Hong Kong 999077, China; 4Department of Electrical Engineering, The Hong Kong Polytechnic University, Hong Kong 999077, China

**Keywords:** smart textiles, fiber optic sensor, inlay, knitted undergarments, FBG, health monitoring, scoliosis

## Abstract

A number of textile-based fiber optic sensors have recently been proposed for the continuous monitoring of vital signs. However, some of these sensors are likely unsuitable for conducting direct measurements on the torso as they lack elasticity and are inconvenient. This project provides a novel method for creating a force-sensing smart textile by inlaying four silicone-embedded fiber Bragg grating sensors into a knitted undergarment. The applied force was determined within 3 N after transferring the Bragg wavelength. The results show that the sensors embedded in the silicone membranes achieved enhanced sensitivity to force, as well as flexibility and softness. Additionally, by assessing the degree of FBG response to a range of standardized forces, the linearity (R^2^) between the shift in the Bragg wavelength and force was found to be above 0.95, with an ICC of 0.97, when tested on a soft surface. Furthermore, the real-time data acquisition could facilitate the adjustment and monitoring of force during the fitting processes, such as in bracing treatment for adolescent idiopathic scoliosis patients. Nevertheless, the optimal bracing pressure has not yet been standardized. This proposed method could help orthotists to adjust the tightness of brace straps and the location of padding in a more scientific and straightforward way. The output of this project could be further extended to determine ideal bracing pressure levels.

## 1. Introduction

Monitoring force and pressure is critical for effective compression therapy, such as bracing, which is the most prevalent type of treatment for Risser stage 0–2 adolescent idiopathic scoliosis (AIS) patients with a Cobb’s angle >25°, aiming to correct their spinal abnormality by passively exerting pressure [1,2,3,4,5]. AIS patients are prescribed a rigid brace, which they must wear for at least 18 h each day [3,6] as tightly as they can tolerate, because the tightness and effectiveness of the brace are positively correlated [7,8]. However, no consensus has been reached on the optimal bracing pressure [9], so the applied pressure mainly depends on the professional judgement of the orthotist [10]. Additionally, many studies have proven that braces have a positive effect on trunk correction, but the mechanism behind this is still unclear [11]. Not only could this knowledge gap result in physical injury, but excessive pressure also affects compliance with bracing and quality of life, thus lowering treatment compliance and efficacy [4]. For example, blood circulation, skin metabolism, and subcutaneous tissues could be affected if the pressure exceeds the typical capillary blood pressure (4.3 kPa) for an extended period [12]. To tackle these issues, a more scientific approach should be developed to prevent excessive pressure by monitoring the forces applied by braces.

A variety of sensors are available to measure the force and pressure exerted by braces, but they all present limitations. For instance, Lou et al. [13] designed a battery-powered wireless force sensor to measure the force exerted by a Boston brace up to 6.78 N (60 kPa). However, some loggers showed a time delay, and the sensors could only be embedded in tailor-made engraved braces. Krištof et al. [14] used a Pressurex^®^ ultra-low pressure film to measure the relative pressure between an AIS patient and a Boston brace, and the maximum recorded pressure was 361.87 kPa. Although they obtained the overall pressure distribution, the exact amount of pressure on the film had to be further analyzed using the Topaq^®^ imaging system and a flatbed scanner that was specifically adapted to the study [15]. Furthermore, attaching a large piece of pressure film on the inside of the brace could cause the film to pleat easily, which may affect the pressure distribution profile. Both types of method were able to measure pressure, but no method has been developed for determining the real-time pressure exerted by braces. Additionally, these two sensors could not easily be applied to the torso and needed to be embedded or cut beforehand. To overcome these challenges, Chan et al. [16] used a Pliance^®^-xf-16 system with 3 × 3 socket sensors to measure the pressure applied by a semi-rigid textile brace, with a pressure range of 0 to 200 kPa. While this was a real-time flexible pressure sensor, it would be difficult to insert the long-tailed sensors into a tightly fitting brace, especially as tape was needed to fix the sensor head to a specific location. Therefore, a flexible and user-friendly sensor with a dynamic pressure profile would be more suitable for measuring bracing pressure. 

Smart textiles that use fiber optic sensors are one of the most appropriate means of obtaining close-to-body measurements, such as interface pressure, because they are very flexible and lightweight and have electromagnetic immunity while maintaining high strain sensitivity [17,18,19]. Furthermore, they make up for the shortcomings of conductive sensors. For example, strain fiber optic sensors can be used during magnetic resonance imaging (MRI) to monitor real-time respiratory activity [20], while conductive sensors cannot. Additionally, fiber optic force sensors can be employed to monitor the blood flow of hospitalized patients who are immobile to prevent decubitus ulcers [21,22,23], examine plantar pressure and gait movement [24,25], and detect hand and joint movement [26,27]. The collected data are useful for detecting diseases and determining the required degree of rehabilitation or the level of athletic performance [28]. 

To address the inadequacies of conventional sensors, we propose a smart textile capable of conveniently measuring bracing pressure. This smart textile is not only easy to use, but it can also accommodate adolescents of different body shapes, so no procedures are necessary before treatment. The smart textile also provides real-time signal profiles and overall pressure distribution. This smart textile was constructed by inserting fiber Bragg grating (FBG) sensors into a knitted undergarment via the inlay method. Additionally, the FBG sensing points were embedded into a thin silicone membrane with a curvilinear shape to enhance the fiber flexibility, robustness, and compatibility of the sensor with the surface of the skin. The smart textile can be applied directly to the body, and the tightly fitting design can prevent the formation of pleats or damage to the fiber optic sensors under compression textile or braces. The real-time data acquisition allows the tightness to be adjusted and monitored during the fitting process.

## 2. Methods

### 2.1. Construction of Smart Textile: Inlaying FBG Sensor into a Highly Elastic Knitted Undergarment

A perfectly fitting smart textile is desirable because pleat formation and optical fiber breakage would be less likely. Therefore, pure PIMA cotton thread (4/80 Nm) was weft knitted using a Shima Seiki knitting machine with a single jersey structure to ensure that the weft direction of the textile had high elasticity (Figure 1a). Then, optical fibers with 4 FBG sensors (wavelengths of 1550.09 nm, 1551.87 nm, 1554.12 nm, and 1555.90 nm, respectively) were warp-inlaid into the knitted undergarment manually to maintain widthwise elasticity (Figure 1b). Hence, the optical fiber was laid straight between the front and back knit loops, while the FBG area was exposed on the fabric surface for embedding into a silicone membrane (Figure 1c). The sensing points were located in the lumbar region, which is one of the main areas subjected to pressure during bracing (Figure 1d).

### 2.2. Embedding FBG Sensors into Silicone Membranes

Embedding the FBG sensors into flexible silicone membranes not only enhanced their flexibility, softness, and susceptibility to compression, but also protected them more effectively. The molding method of Guo et al. [29] comprised the embedding of FBG sensors into a flexible polydimethylsiloxane (PDMS) membrane with a curvilinear shape. This improved the elasticity of the fibers during fabric stretching, compression, and twisting, as well as the strain sensitivity. However, Dragon Skin^®^20 silicone was used instead of a polydimethylsiloxane (PDMS) membrane in this experiment because it does not require high-temperature curing, but it could potentially be integrated into textiles. For example, Presti et al. [30] encapsulated a sensor in Dragon Skin^®^20 silicone rubber and integrated this into an elastic band to monitor respiratory activity and heartbeat. This improved not only the sensor robustness and the compatibility of the sensor and the skin, but also the sensitivity of the physiological parameters. 

For the embedding process, a mold with a depth of 1.5 mm was first prepared. Then, a bent optical fiber was placed in the middle of the mold, which was subsequently filled with Dragon Skin^®^20 silicone (Figure 2a). This was intended to create a groove in the silicone. After the silicone was dried at room temperature for 75 min, it was carefully peeled off to reveal a groove (Figure 2b). The optical fiber was warp-inlaid into the knitted undergarment, but the FBG sensing area was exposed on the fabric surface for embedding (Figure 2c). Then, the FBG sensors were lifted from the knitted undergarment and embedded into this groove (Figure 2d). Finally, the embedded FBG sensors were covered with silicone to secure them into position (Figure 2e). Figure 2f shows a cross-section of an embedded FBG sensor warp-inlaid into the knitted structure. Aside from enhancing the sensor elasticity and sensitivity, the embedding of the FBG sensors also reduced the tendency of the sensors to shift. To integrate four FBG sensors into the silicone membranes, this method was performed four times.

### 2.3. Wear Trial to Simulate Bracing

#### 2.3.1. Smart Textile

The applied forces could not be obtained by directly measuring the peak Bragg wavelength shifts, so it was necessary to understand the force–strain relationship. An experiment that simulated a bracing treatment was conducted to understand the relationship between the normal forces and strain sensed by the FBG sensors. In the experiment, artificial skin and a force gauge (model: JSV-H1000, Japan Instrumentation System) were used to model human skin and bracing pressure, respectively. The artificial skin was 15 mm thick and made of Ecoflex^®^ 0010 silicone rubber, which has a similar texture to the human torso. Chan [31] indicated that the Young’s modulus and Poisson’s ratio of a soft mannequin made of Ecoflex^®^ 0010 are 0.0768 MPa and 0.3, respectively. Furthermore, the pressure at the interface of a brace and a human torso is similar to that at the interface of a brace and a soft mannequin. Moreover, they have similar spinal corrective effects and were validated using a finite element model. The embedded FBG sensors were inserted into the knitted garment and placed between the force gauge and the artificial skin; see Figure 3. Every 5 s, the force gauge exerted a force on the smart textile that rose by increments of 0.5 N until it reached 10 N. 

FBG sensors are sensitive to strain, which can be induced by axial and transverse forces. Typically, the strain of optical fibers is defined as axial force because they show prominent deformation. For example, FBG sensors are lengthened when the fibers are stretched. However, transverse forces mainly change the fiber cross-section and slightly lengthen the fibers because of the positive Poisson’s ratio of the fibers. Fixing the two ends of the FBG sensor could improve the transverse force sensitivity [29,32]. When the two ends of a flexible string are fixed and transverse force is applied to the string, higher axial forces result; see Figure 4. This also implies that the corresponding deformation on the optical fiber caused by transverse forces will eventually convert to axial strain, which can be easily measured by calculating the peak Bragg wavelength shift. Therefore, the relationship between force and Bragg wavelength can be established, and we predicted that a higher peak Bragg wavelength would shift to a higher force.

#### 2.3.2. Pliance^®^-X Pressure Sensor

The Pliance^®^-X pressure sensor was compared with conventional pressure sensors, and used for a wear trial simulation. The sensor was tested under two conditions: (1) placed between two glass clips to simulate the calibration system (Figure 5a), and (2) placed on the artificial skin (Figure 5b). The Pliance^®^-X pressure sensor is a reliable tool for measuring low interface pressure between textiles and skin under stationary conditions [33,34,35]. Its very small and ultra-thin sensing area is connected to an extended conductive strip, which makes it favorable for insertion underneath long-sleeved clothing. Moreover, due to the flexible sensor configuration, a single sensor or a matrix of sensors can be used for various measurements. Its high flexibility has captured much research interest, and studies have used this device to measure bracing pressure. For example, the Pliance^®^-X pressure sensor was used to measure the amount of pressure exerted by a posture correction girdle on AIS patients [16] and a functional brace on anterior-cruciate-ligament-deficient (ACLD) subjects [34].

### 2.4. Data Analysis

The data collected from the wear trials were used to examine the linearity (R^2^) of the smart textile by assessing the degree to which the FBG sensors responded to a range of standardized forces. The force gauge applied a loading force of 0.5 N every 5 s onto the smart textile until it reached 10 N, and 10 readings were recorded every second. In total, 50 readings were taken every 5 s, but only the peak Bragg wavelength shifts were extracted to analyze the linearities of the FBG sensors at 1550.09 nm (FBG 1), 1551.87 nm (FBG 2), 1554.12 nm (FBG 3), and 1555.90 nm (FBG 4). Furthermore, the linearity was compared to that of the Pliance^®^-X pressure sensor when tested on a soft surface. Moreover, to evaluate the test–retest reliability of the smart textile, a two-way random effects model intraclass correlation coefficient (ICC) with 95% confidence intervals was used.

## 3. Results and Discussion

### 3.1. Smart Textile Design

The knitted undergarment showed high elasticity and fit the dummy perfectly. The waistline was shaped, and an open chest design was adopted to prevent issues with the opening and compression (Figure 6). In addition, no side seam was used, thus maximizing wear comfort. To avoid rolling edges, purl knit and 2 × 2 rib were applied for the strap edges and hemline, respectively. Then, optical fibers with four FBG sensors were inlaid manually in the warp direction into the knitted undergarment at the lumbar region, which is one of the main areas subjected to pressure during bracing. 

Typically, straight configurations of optical fibers in a knitted structure reduce the fabric elasticity in the direction of insertion [36]. Bending the optical fibers into a curvilinear shape along the warp direction solved this problem and preserved the widthwise elasticity of the fabric. Moreover, the sensor flexibility and softness were improved after the FBG sensors were embedded into the silicone membranes in a curvilinear shape. 

### 3.2. Linearity of Silicone-Embedded FBG Sensors and Bare FBG Sensors 

The results showed that the silicone-embedded FBG sensors were more stable and linear than the bare FBG sensors when force was applied. The Bragg wavelength delta difference indicated the force sensing range and force sensitivity; a larger delta difference denoted a higher sensitivity. For the embedded FBG sensors, the Bragg wavelength delta differences from 0 N to 10 N of FBG sensors 1 to 4 were 0.157 nm, 0.221 nm, 0.236 nm, and 0.175 nm, respectively (Figure 7a), while those for the bare FBG sensors were 0.196 nm, 0.253 nm, 0.233 nm, and 0.169 nm, respectively (Figure 7b). Although the bare FBG sensors had a higher sensing range, their linearity and stability were inferior to those of the embedded FBG sensors. For example, the R^2^ values of the silicone-embedded and bare FBG sensors were 0.90-0.92. and 0.76–0.89, respectively. Their equations are:Embedded FBG 1: y = 0.0145x + 0.0283(1)
Embedded FBG 2: y = 0.0208x + 0.0422(2)
Embedded FBG 3: y = 0.0216x + 0.0513(3)
Embedded FBG 4: y = 0.0158x + 0.0386(4)
Bare FBG 1: y = 0.0201x + 0.0321(5)
Bare FBG 2: y = 0.0255x + 0.0433(6)
Bare FBG 3: y = 0.0223x + 0.0530(7)
Bare FBG 4: y = 0.0153x + 0.0499(8)

One of the reasons for this may be that the two ends of the FBG sensors were fixed in the silicone membranes, which simultaneously minimized the shifting of the sensors and maximized their sensitivity to the transverse force. Therefore, the signal could be transferred more efficiently and stably so that the silicone-embedded FBG sensors showed better linearity. Another reason for this could be the sensor flexibility, which was enhanced by its curvilinear shape and the silicone membrane. The former allowed the optical fiber to bend more without breaking, while the latter increased the compression susceptibility. A third possible reason is that the flat silicone membranes increase the sensing area and distributed the force more evenly. The FBG sensors were tiny points (125 μm in diameter) on the optical fiber, so a minor displacement of the force gauge could have affected data acquirement. However, when the FBG sensors were embedded into the silicone membrane, the force applied near the FBG sensors could also be detected, which minimized the percentage of error. The pre-bent FBG sensors embedded into the silicone membranes eliminated shifting of the sensor and enhanced the sensor flexibility and compression resistance, as well as accuracy.

### 3.3. Linearity and Reliability of Silicone-Embedded FBG Sensors and Pliance^®^-X Pressure Sensor

The Pliance^®^-X pressure sensor showed high linearity for both tests, with an R^2^ value above 0.97 (Figure 8a,b). However, its performance on hard and soft surfaces varied greatly. When tested between two glass clips, the detected forces were nearly identical to that of the force gauge, but their differences were more substantial when tested on artificial skin with increases in force. For example, when the force gauge applied a loading force of 1.5 N, the detected force was only 0.6 N, which implied that the force was absorbed by the artificial skin, and the sensor may not be suitable for use on a very soft surface with a high compression force. Moreover, the Pliance^®^-X sensor was only suitable for measuring forces up to 2.5 N (~30 kPa), while the FBG sensors could withstand a loading force of more than 10 N. This agreed with Lai and Tsang [33], who found that the Pliance^®^-X pressure sensor is a reliable tool when measuring low interface pressure. The experimental findings showed that the Pliance^®^-X pressure sensor had two main limitations, namely, low accuracy on soft surfaces and the inability to measure a high interface pressure. 

To compensate for the shortcomings of the Pliance^®^-X sensor, the smart textile was designed to measure the force on a soft elastic surface, such as human skin. All of the data were collected on soft artificial skin (with a Young’s modulus of 0.0768 MPa), and the detected force was measured using the force gauge. When the force applied ranged from 0 N to 10 N, all of the linearities between the peak-shifted Bragg wavelengths and forces of FBG sensors 1 to 4 showed a highly positive correlation, with an R^2^ of 0.90–0.92. When the scale of the force was 3 N, the R^2^ value was 0.95–0.97 (Figure 9), which approximates that of the Pliance^®^-X sensor when tested on soft material (R^2^ = 0.9971) (Figure 8b). Moreover, the FBG sensors had an intraclass correlation coefficient (ICC) of 0.97, which means that the smart textile was highly reliable when tested on a soft surface. Substituting the Bragg wavelength delta difference (x) into the equation, the corresponding force (y) within 3 N could be found as follows.
FBG 1: y = 0.0289x + 0.0070(9)
FBG 2: y = 0.0424x + 0.0099(10)
FBG 3: y = 0.0458x + 0.0157(11)
FBG 4: y = 0.0334x + 0.0131(12)

#### Challenges of Conventional Integration Methods

Besides their sensing abilities, compatibility between the sensors and textiles may be the key to the success of smart textiles, because this factor greatly affects the final performance factors, such as flexibility and wear comfort. There are five main methods of integrating fiber optic sensors into textiles, namely, adhering; sewing; embroidering; embedding; and inserting (e.g., weaving and knitting) them [28]. In the first generation of smart textiles that used fiber optic sensors, the sensing components were adhered and sewn onto the textile surface. Due to the simplicity of this method, the sensing component could be easily detached, which worked well for ongoing maintenance and laundering. For example, Grillet et al. [37] sewed and glued fiber optic sensors onto elastic bandages in a curvilinear layout and applied them to the abdomen region to monitor respiratory movement during MRIs. However, the protruding sensors may affect the aesthetics of the product. 

The second generation of smart textiles used fiber optic sensors as textile yarn, i.e., the sensors were embroidered onto the textile surface, or woven, and knitted into the fabric. Similar to sewing, embroidering fixes the fiber optic sensors onto the fabric surface, but no sewing thread is used. This is because the fiber optic sensors can be directly used as thread and bent into a specific shape. For instance, Quandt et al. [21] embroidered two rows of fiber optic sensors on a bed sheet in continuous loops to monitor heartbeat and blood flow for examining wound healing and preventing decubitus ulcers. However, the excessive bending of these fibers may affect signal transmission and even break the optical fibers. Inserting optical fibers into a textile structure seems to be a more effective solution, because the sensors are fully hidden in the fabric while maintaining high functionality. However, this method could be time-consuming, because the fabric structure and material properties must be considered before inserting the sensors. For example, Rothmaier et al. [38] inserted a polymer fiber optic sensor into woven fabric with a twill structure to measure the applied compression forces. One of the most important factors may be the strength of the optical fibers, because they are subjected to a high degree of tension during the weaving process. Weak fibers may break or deform easily, and thus, lose their sensing ability. Knitting is an effective choice if elastic textiles are required, and the fiber optic sensors can be used in a similar manner to inlaid yarn, i.e., they can be laid straight between the front and back knit loops. Koyama et al. [39] inserted silk-covered FBG sensors into a knitted wristband to monitor blood pressure. Although the silk-covered optical fibers contributed to the softness of the wristband, its elasticity could have been degraded, as the optical fibers had low stretchability. 

The third generation of smart textiles focused on enhancing their flexibility. One of the common methods of embedding fiber optic sensors into polymer substrates is to bend them into curved lines. Not only can this protect the sensors, but they are also made more robust and compatible with skin [29,30]. For instance, an ultra-flexible skin-like FBG strain sensor was developed by embedding curvilinear FBG sensors into a thin PDMS membrane, and then, attaching the membrane onto kneepads and gloves [29]. As these items are lightweight and flexible, they do not inhibit mobility, which is useful for measuring movement. Furthermore, a 3D-printed insole was developed by embedding 15 macro-bending sensors into an insole in a curvilinear shape; this was used to monitor plantar pressure and gait movement [24]. Although the disadvantage of embedding is the increased bulkiness of the smart textile, this method is ideal for creating smart textiles that are subjected to compression forces due to its robustness to higher compression forces. 

Our novel method for manufacturing a force-sensing smart textile combined the advantages of the inlay and embedding techniques. The inlay approach fully concealed the fine optical fibers in the smart textile while maintaining the wear comfort of the material. Additionally, inserting optical fibers manually ensured that the FBG sensors were not overly stretched and prevented breakage during the manufacturing process. Moreover, the location of the FBG sensors could be determined without complicated calculations, therefore ensuring time-efficiency. Additionally, embedding FBG sensors into silicone membranes with a curvilinear shape enhanced their flexibility, provided extra protection, and improved compatibility while maintaining sensitivity to force. However, the rigidity of optical fibers and textile yarn varies, so their integration would affect the softness and elasticity of the knitted undergarment to differing extents.

## 4. Limitations of Experiments and Future Works

Below are listed serval limitations of the current study. The results obtained using a soft surface demonstrated lower robustness than those obtained using a hard surface due to the cushioning effect, which indicates that the energy was absorbed. For example, the reliability of the Pliance^®^-X pressure sensor was assessed in [35], where the authors reported an ICC of 0.998 on a hard surface, and an ICC of 0.87 on children’s skin. This implied that the softness of the tested material may affect the sensor’s reliability. In this project, the artificial skin was as soft and elastic as a human torso, so the energy was also absorbed. Although a force of 0.5 N was applied every 5 s to the smart textile and the Pliance^®^-X pressure sensor, the mean detected forces were approximately 0.45 N and 0.2 N, respectively. Therefore, the embedded FBG sensors took longer than the bare FBG sensors to reach 10 N (120 s and 105 s, respectively). Nevertheless, their sensitivity to force was not greatly affected. For example, the Bragg wavelength delta differences in the embedded and bare FBG sensors at 1552 nm were 0.221 nm and 0.253 nm, respectively. The overall results indicated that the bare FBG sensors achieved higher sensitivity to force but tended to become more stable and linear when embedded in silicone membranes. 

Besides the deformation caused by applied force, the shifted Bragg wavelength was also affected by temperature. To minimize the error caused by body temperature, polarization-maintaining fibers could be used to discriminate strain and temperature, for example, highly birefringent fiber Bragg grating [40] and polarization Brillouin reflectometry [41]. Furthermore, the shifted Bragg wavelength could be affected by the deformation of the FBG sensors, as well as the temperature. The long tail of the optical fiber that connected to the heavy interrogator (Micron Optics SM130) may affect the wearability of the smart textile, so AIS subjects should wear these cautiously. Developing a mini-interrogator would be the next step in enhancing the wearability of this smart textile, making it portable and more user-friendly.

## 5. Conclusions

We introduced a novel method for manufacturing a force-sensing smart textile by inlaying FBG sensors into a knitted structure along the warp direction and embedding FBG sensors into silicone membranes. The results showed that the silicone-embedded FBG sensors were more stable and linear than the bare FBG sensors when force was applied. Not only was the sensor flexibility enhanced, but the embedded FBG sensors eliminated the tendency of the sensor to shift, and enhanced compressional susceptibility and accuracy. When the force range was 0–10 N, the linearity (R^2^) between the Bragg wavelengths and forces was above 0.90. However, the R^2^ (>0.95) value was even higher when it was reduced to 3 N, which was comparable to the Pliance^®^-X sensor when evaluating a soft material. Furthermore, the FBG sensors had an ICC of 0.97, indicating that the smart fabric was highly reliable when evaluated on a soft surface. Since no consensus has been reached on the optimal bracing pressure, smart textiles could serve as a useful tool for tracking the forces exerted by a brace during the fitting process. Smart textiles could make the process of adjusting the brace simpler and more scientific for orthotists. Our results could also be used to establish optimal bracing pressure levels, thus preventing AIS patients from having to endure excessive pressure, while maintaining an appropriate amount of bracing force in order to halt curve progression, and reduce the risk of pressure-related injuries.

## Figures and Tables

**Figure 1 sensors-23-05145-f001:**
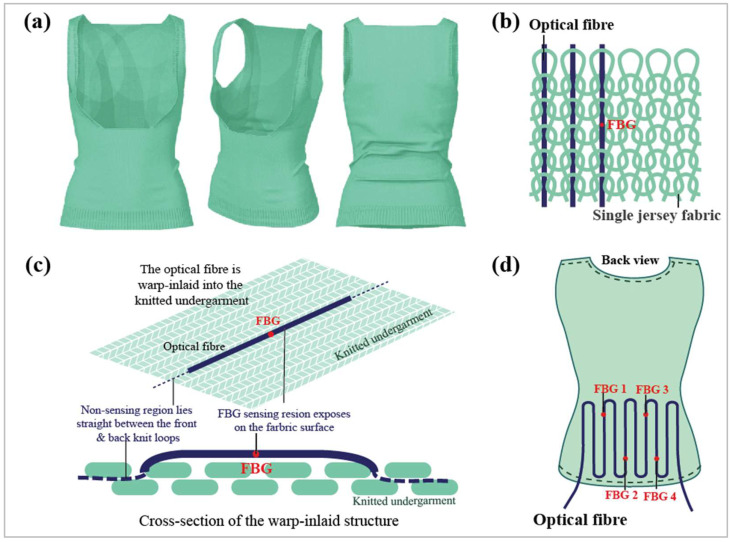
(**a**) Three-dimensional illustration of smart textile, (**b**) structure of optical fiber warp-inlaid into the single jersey fabric, (**c**) cross-section of the warp-inlaid structure, and (**d**) configuration of the optical fiber and allocation of FBG 1 to 4.

**Figure 2 sensors-23-05145-f002:**
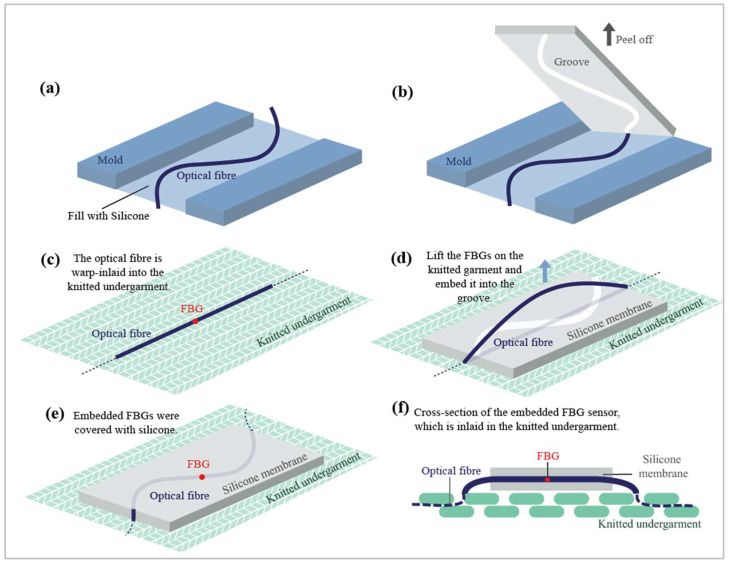
Schematic of embedding FBG sensors into a silicone membrane: The process entails (**a**) filling the mold with silicone, (**b**) peeling off the resulting silicone membrane, (**c**) exposing the FBG sensing area for embedding, (**d**) embedding the FBG sensor into the groove of the silicone membrane, (**e**) covering the embedded FBG sensor with silicone, and (**f**) showcasing a cross-sectional view of an embedded FBG sensor that is warp-inlaid into the knitted structure.

**Figure 3 sensors-23-05145-f003:**
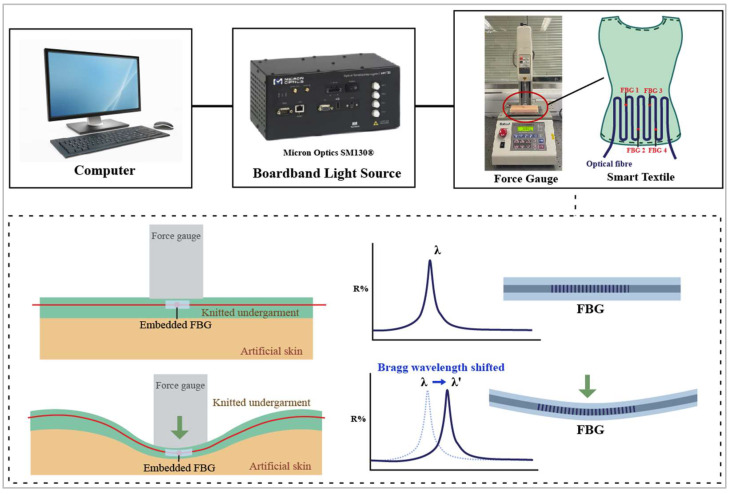
Equipment setup and schematic of transverse force applied to smart textile on artificial skin.

**Figure 4 sensors-23-05145-f004:**
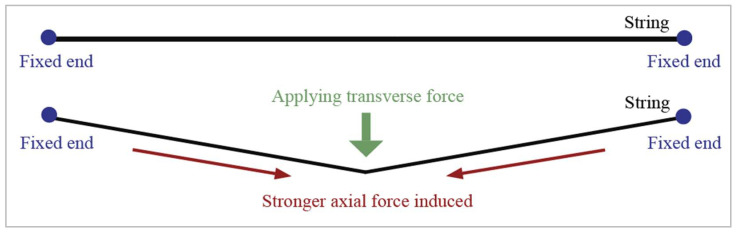
Schematic of amplifying string axial and transverse forces.

**Figure 5 sensors-23-05145-f005:**
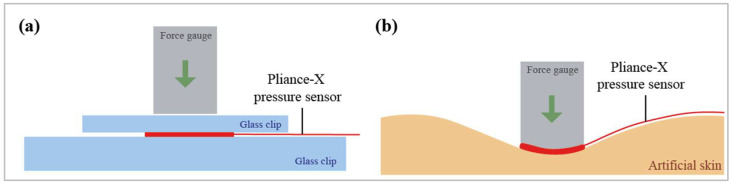
Schematic of transverse force applied to Pliance^®^-X pressure sensor: (**a**) between two glass clips and (**b**) on artificial skin.

**Figure 6 sensors-23-05145-f006:**
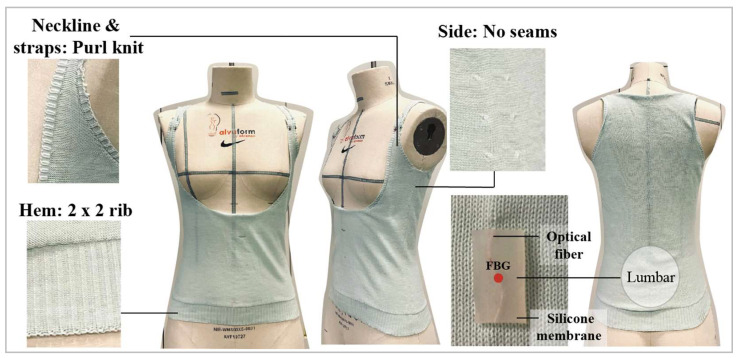
Seamless knitted undergarment created using Shima Seki knitting machine, and silicone-embedded FBG sensor inlaid in the warp direction.

**Figure 7 sensors-23-05145-f007:**
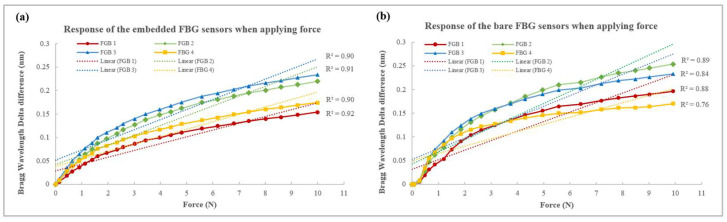
Linearity of Bragg wavelength when applying force on (**a**) silicone-embedded FBG sensors, and (**b**) bare FBG sensors.

**Figure 8 sensors-23-05145-f008:**
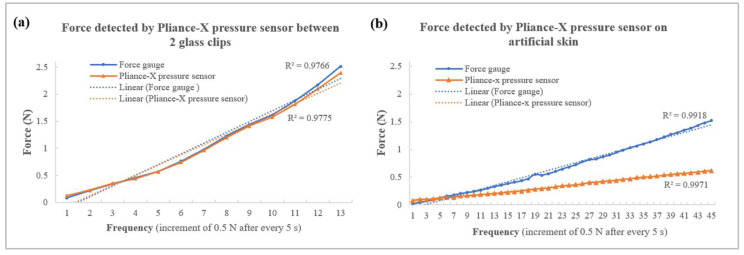
Force detected by Pliance^®^-X pressure sensor (**a**) between two glass clips and (**b**) on artificial skin.

**Figure 9 sensors-23-05145-f009:**
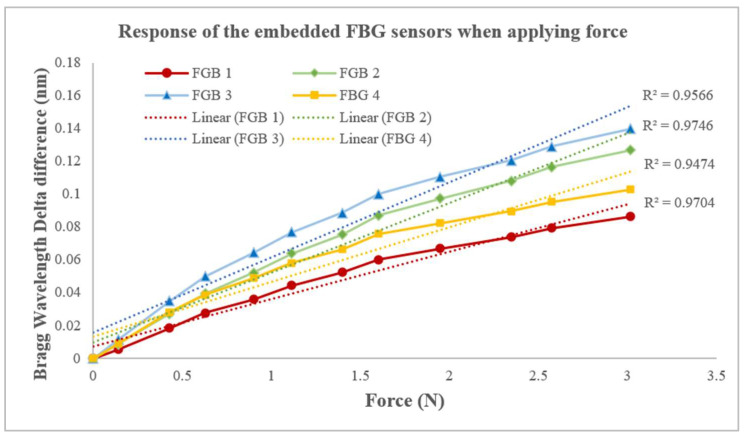
Linear regression between Bragg wavelength delta difference and force when applying force to silicone-embedded FBG sensors 1 to 4.

## Data Availability

The datasets that were generated and/or analyzed during the current study are available upon reasonable request from the corresponding author.

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
