# Peer review of "A Novel Force-Sensing Smart Textile: Inserting Silicone-Embedded FBG Sensors into a Knitted Undergarment"

_sensors, 2023, doi:10.3390/s23115145_

Round 1
Reviewer 1 Report
The work presented by the authors is a complete scientific study, important from a practical point of view. The scientific soundness of the title and the article itself is quite high. The disadvantages of the work include a slightly confusing narration, the presence of many typos and layout errors.
I would like to draw the attention of the authors to the following remarks:
- Line 28: Please remove "Correspondence: joanne.yip@polyu.edu.hk; Tel.: +852-2766-4848" from this line;
- Line 30: Why do you start from the Ref [13]? And were are the links to the Refs [1-12]? It looks like a huge part of an introduction is missed, please bring it back;
- Line 65: The authors start working with FBGs with no comparison with other approaches. Could the optical reflectometry in frequency domain (OFDR) provide more detailed data?;
- Line 84: Is this possible to show the FBGs positions on the Figure 1?;
- Line 97: The start of the sentence sounds strange in English, please rephrase;
- Line 113: The figure 2 and its description above do not represent the process clearly. Do this picture's part (a-f) mean different stages or different versions of FBGs embedding? It is not clearly seen until you read the whole paper;
- Line 192 abd 193: Please remove strage symbols from these lines;
- Line 203: The section ends with the Figure. Please place this after the first mentioning in text;
- Line 218: The Figure 7 where we see the wavelength difference versus time has no physica meaning. I think this would be great if you label the x-axis in Newtons.
- Line 240: The section ends with the Figure again. Please place this after the first mentioning in text;
- the term "linearity" is constantly used in the work, but none of the figures has a strictly linear shape! I would like to ask the authors to highlight (graphically) those parts of the curves that they consider to be linear;
- An important issue that was left out of the scope, is the discrimination of temperatures and deformations in the sensor. It is known fact that different locations of the human body have different temperatures, while non-stationary in time. It doesn’t matter which sensor is used: distributed or based on FBGs, you can separate the influence of two factors using PM fiber [1-3]. Perhaps this should be considered as future work and mentioned in the Conclusions section;
- One more thing that needs clarification. What are the dimensions of the interrogator and its cost? How comfortable and economically feasible is wearing such clothes?;
- (just as a proposal) The paper lacks the experimental setup figure with optical scheme (if applicable).
[1] DOI: 10.1364/OFS.2006.ThC5
[2] DOI: 10.1109/JLT.2019.2959671
[3] DOI: 10.1134/S0020441220040223
I think this is not ideal, but acceptable, and a lot of typos need to be fixed.
Author Response
Date: 17th May 2023
Re: sensors-2385279. R1 “A novel method to create force-sensing smart textile: Inserting silicone-embedded FBG sensors into knitted undergarment"
Thank you very much for considering our manuscript entitled “A novel method to create force-sensing smart textile: Inserting silicone-embedded FBG sensors into knitted undergarment”. We appreciate the valuable comments from editors, which not only assist us to improve the quality of our manuscript but also provide some creative ideas for future studies. We have carefully revised the manuscript according to the specific comments, and corresponding changes are listed below:
The specific changes made in response to the comments of Reviewer 1 are as follows.
Point 1: Line 28: Please remove "Correspondence: joanne.yip@polyu.edu.hk; Tel.: +852-2766-4848" from this line;
ANS 1: Thank you for your reminders
Thank you for bringing this to our attention. We have taken note of your reminder and have removed Line 28 from the manuscript as per your suggestion. (Line 49)
Point 2: Line 30: Why do you start from the Ref [13]? And where are the links to the Refs [1-12]? It looks like a huge part of an introduction is missed, please bring it back;
ANS 2: I humbly admit that I made some mistakes because of overlooking the update of the Endnote, and I am striving to correct them to the best of my ability. The deleted paragraph is mentioned in the Introduction. All in-text citations have been edited to match the reference list, thank you for your reminders and suggestion. (Line 33-48)
Point 3: Line 65: The authors start working with FBGs with no comparison with other approaches. Could the optical reflectometry in frequency domain (OFDR) provide more detailed data?
ANS 3: Thank you for your question. We chose FBG as our sensor because it is one of the most popular sensors used in textile integration [1]. Additionally, previous studies have shown that FBGs can be embedded with a polymer membrane, making it a suitable choice for our project [2].
The aim of our project is to demonstrate the feasibility of warp-inlaying optical fiber into textiles and proposing a method to increase the flexibility and sensitivity of FBG as a force sensor. While OFDR (distributed sensor) can provide more detailed data, it is a more expensive option. We believe that measuring the signal at a specific position by FBG (quasi-distributed) is sufficient for the purpose of creating the smart textile. We appreciate your interest in our work and hope that this explanation provides a better understanding of our sensor selection. If you have any further questions or comments, please do not hesitate to let us know. Thank you.
Point 4: Line 84: Is this possible to show the FBGs positions in the Figure 1?
ANS 4: Thank you for your suggestion. We agree that adding the positions of the FBG sensors will enhance the understanding of our application. Therefore, we have added the configuration of the optical fiber and the allocation of FBG sensors in Manuscript_v2_Figure 1d. Additionally, we have included a cross-section of the warp-inlaid structure in Manuscript_v2_Figure 1c to show the structure before embedding FBGs into the silicone membranes. We hope that these additions provide greater clarity and understanding of our work.
Point 5: Line 97: The start of the sentence sounds strange in English, please rephrase;
ANS 5: Thank you for your reminder, the sentence has been rephrased. (Line 124-128)
Point 6: Line 113: The figure 2 and its description above do not represent the process clearly. Do this picture's part (a-f) mean different stages or different versions of FBGs embedding? It is not clearly seen until you read the whole paper;
ANS 6: We apologize for any confusion caused by Manuscript_v1_Figure 2 and its description. To clarify the embedding process, we added a sentence “To integrate four FBG sensors into the silicone membranes, the method was performed four times.”. Additionally, we modified Manuscript_v2_Figure 2d and f to better illustrate how to embed the FBGs on the knitted undergarment surface. (Line 139-143, 145-146)
Point 7: Line 192 and 193: Please remove strange symbols from these lines;
ANS 7: Thank you for your reminder, the symbols “¬” are removed. (Line 234-235)
Point 8: Line 203: The section ends with the Figure. Please place this after the first mention in text;
ANS 8: Thank you for your suggestions, Manuscript_v1_Figure 6 is moved after the first mentioning in the text. (Line 239-242)
Additionally, we have relocated the 3D illustration of the smart textile (Manuscript_v1_Figure 6a) and the configuration of the optical fiber and FBG sensors (Manuscript_v1_Figure 6c) to Manuscript_v2_Figure 1a and 1d, respectively. This is because they were first mentioned in section 2.1 and relocating them makes the manufacturing process of the force-sensing smart textile clearer (Line 112-117).
Point 9: Line 218: The Figure 7 where we see the wavelength difference versus time has no physical meaning. I think this would be great if you label the x-axis in Newtons.
ANS 9: Thank you for your kind suggestion regarding our manuscript. We appreciate your attention to detail and your efforts to help us improve the scientific quality of our work.After careful consideration, we agree that changing the x-axis from time frame to force (N) would be more scientifically accurate. Therefore, we have decided to change the x-axis in Manuscript_v1_Figure 7 to force (N) as well, in order to maintain consistency with Manuscript_v1_Figure 8. However, we have found that after changing the x-axis, the two figures look quite similar. Therefore, we have decided to combine them in order to better illustrate the Bragg wavelength delta difference and linearity. (Line 277-280)
Regarding Manuscript_v2_Figure 7, we have reduced the number of data points on the x-axis to around 20, as we found that only the peak shifted Bragg wavelength against force could be accurately extracted. This is because the response of FBG sensors is faster than the force gauge, making it difficult to synchronize the fluctuating force with the shifted Bragg wavelength. However, we have found that the rapid increase of the shifted Bragg wavelength (peak shifted wavelength) and force can still be easily identified. We would also like to clarify that the fluctuating force observed in our experiments is caused by the energy absorption of soft materials. For example, although 0.5 N was applied every 5 s to the FBG sensors, the mean detected force was approximately 0.45 N only.
Point 10: Line 240: The section ends with the Figure again. Please place this after the first mentioning in text.
ANS 10: Thank you for your suggestion, Manuscript_v1_Figure 8 is modified as Manuscript_v2_Figure 7 now, and it is moved after the first mentioning in the text. (Line 277-280)
Point 11: the term "linearity" is constantly used in the work, but none of the figures has a strictly linear shape! I would like to ask the authors to highlight (graphically) those parts of the curves that they consider to be linear;
ANS 11: We apologize for any confusion caused by the figures in our manuscript. We understand that it can be difficult to compare their performance without the linear lines. To address this concern, we have included all linear lines in both Manuscript_v2_Figure 7 and 8 (Line 277-280, 319-324).
We believe that this modification will enhance the clarity and scientific quality of our work and enable readers to better understand our findings.
Point 12: An important issue that was left out of the scope, is the discrimination of temperatures and deformations in the sensor. It is known fact that different locations of the human body have different temperatures, while non-stationary in time. It doesn’t matter which sensor is used: distributed or based on FBGs, you can separate the influence of two factors using PM fiber [1-3]. Perhaps this should be considered as future work and mentioned in the Conclusions section;
ANS 12: We agree that different temperature and deformation can affect the accuracy of FBG sensors, and we recognize that this is one of the major issues when using them. As such, we have addressed this problem and possible solutions in the Limitations of Experiments and Future Works section of our manuscript. (Line 431-436)
Point 13: One more thing that needs clarification. What are the dimensions of the interrogator and its cost? How comfortable and economically feasible is wearing such clothes?;
ANS 13: Thank you for your question regarding the Micron Optics SM130® interrogator. The cost of this interrogator is approximately USD 30,000, with dimensions of 12 x 27 x 14 cm. We recognize that its cost and weight are major disadvantages of using this interrogator. We understand that portable sensors are the main trend, and that this smart textile may not be wearable like other textile-based fiber optic sensors. The long tail of the optical fiber and heavy interrogator would be major issues when conducting clinical trials. Therefore, our next step is to develop a mini interrogator like the FiSpec FBG X100 [3], which costs USD 200. We have mentioned these limitations and future approaches in the Limitations of Experiments and Future Works section of our manuscript (Line 436-440).
Regarding the choice of material for the smart textile, we have selected pure cotton yarn to knit the textile. This is because pure cotton undergarments are the most commonly worn under braces by AIS patients due to their high comfort and low friction. Furthermore, the seamless design can improve comfort by reducing discomfort caused by seam allowances, and the open chest design can prevent issues with compression and opening.
Point 14: (just as a proposal) The paper lacks the experimental setup figure with optical scheme (if applicable).
ANS 14: Thank you for your suggestion, Manuscript_v1_Figure_3 is edited. (Line 176-178)
Point 15: Comments on the Quality of English Language: I think this is not ideal, but acceptable, and a lot of typos need to be fixed.
ANS 15: Thank you for taking the time to review my manuscript and provide valuable feedback. I appreciate your comments and suggestions, and I have carefully considered them in preparing my revised manuscript. Regarding the language quality, I understand that English is not my native language. I have already submitted my manuscript_v2 to a professional editing service (MDPI English editing) to address any errors. I am committed to improving the language quality of my manuscript and ensuring that it meets the standards of the journal. I hope that my revised manuscript reflects this commitment.
Reference
- Lee, K.-P.; Yip, J.; Yick, K.-L.; Lu, C.; Lo, C.K. Textile-based fiber optic sensors for health monitoring: A systematic and citation network analysis review. Textile Research Journal 2022, 92, 2922-2934, doi:10.1177/00405175211036206.
- Presti, D.L.; Massaroni, C.; D’Abbraccio, J.; Massari, L.; Caponero, M.; Longo, U.G.; Formica, D.; Oddo, C.M.; Schena, E. Wearable system based on flexible FBG for respiratory and cardiac monitoring. IEEE Sensors Journal 2019, 19, 7391-7398.
- FiSens. PRECISE, COMPACT AND COST-EFFECTIVE FiSpec FBG X100. Available online: https://fisens.com/products/fispec-fbg-x100 (accessed on 10 May).

Reviewer 2 Report
The english language must be corrected at a few plades. Some sentences are difficult to understand. For example line 15: force within 3 N???
line 18: it is much better to write: shift of Bragg...
line 28: correspondence adress should be moved.
line 31 force .... 60 Pa. Forces are expressed in N and pressure in Pa.
line 34: 30 psi: in scientific journal only metric units are acceptable
line 130: I am surprised about the very small young modulus 0.07. Is this expressed in Pa, kPa, MPa?????
line 135; this sentence is not understandable!
linearity is sometimes written as R2 and sometimes as R2. I am also very surprised that the parabolic curves in most of the figures give rise to very high values of R2. The authors made then the wrong conclusion that everything is quite linear. It would be useful to mention the equation how the calculation of R2 is done. It is no that difficult to present second order polynomials instead of (1-4).
Nothing is said about how wearable this system is. You do not need just a sensor but also a laser light source and an optical system to measure the shift of the Bragg wavelength. I can hardly imagine a wearable interferometer. Or is this system only useful for people in a hospital who cannot move?
I have no remarks about the research itself. My comments are mainly about the presentation of the research in this manuscript. So I came to the conclusion major reviewing. I will welcome a re-reviewing.
see comments to authors.
Author Response
Date: 17th May 2023
Re: sensors-2385279_R1 “A novel method to create force-sensing smart textile: Inserting silicone-embedded FBG sensors into knitted undergarment"
Thank you very much for considering our manuscript entitled “A novel method to create force-sensing smart textile: Inserting silicone-embedded FBG sensors into knitted undergarment”. We appreciate the valuable comments from editors, which not only assist us to improve the quality of our manuscript but also provide some creative ideas for future studies. We have carefully revised the manuscript according to the specific comments, and corresponding changes are listed below:
The specific changes made in response to the comments of Reviewer 2 are as follows.
Point 1: line 18: it is much better to write: shift of Bragg...
ANS 1: Thank you for your helpful suggestions and reminders regarding our manuscript. We appreciate your attention to detail and your efforts to improve the clarity and accuracy of our work, and the sentence has been modified. (Line 22)
Point 2: line 28: correspondence address should be moved.
ANS 2: Thank you for bringing this to our attention, the correspondence address is deleted. (Line 49)
Point 3: line 31 force .... 60 Pa. Forces are expressed in N and pressure in Pa.
ANS 3: Thank you for your reminder, it should be 6.78 N, and the information is updated. (Line 52)
Point 4: line 34: 30 psi: in scientific journal only metric units are acceptable.
ANS 4: Thank you for your reminder, and the information is updated. (Line 54-56)
Point 5: line 130: I am surprised about the very small young modulus 0.07. Is this expressed in Pa, kPa, MPa?????
ANS 5: Sorry for missing the unit, it should be 0.0768 MPa, and it is modified in the text. (Line 165, 327)
Point 6: line 135; this sentence is not understandable!
ANS 6: We apologize for any confusion caused by the unclear phrasing, and we have revised the sentence accordingly. (Line 170-174)
Point 7: linearity is sometimes written as R2 and sometimes as R2. I am also very surprised that the parabolic curves in most of the figures give rise to very high values of R2. The authors made then the wrong conclusion that everything is quite linear. It would be useful to mention the equation how the calculation of R2 is done. It is not that difficult to present second order polynomials instead of (1-4).
ANS 7: Thank you for your reminder, all “R2” are edited as “R2”. (Line 218, 306, 331-333, 451-452)
We acknowledge your concerns about the presentation of the data in Manuscript v1_Figure 8, and we appreciate your efforts to help us improve it. We have generated Figure 8(2) (FBG sensors separated into different graphs with trendlines) to provide a better representation of the linearity R2 of FBG sensors. Although both of their R2 are generated by the Excel (Format trendline - Linear), the overlapping lines in Manuscript v1_Figure 8 seem to cause an illusion that the lines are more curved than they should be. Also, we double-checked the data by using SPSS, the Pearson correlation (between force and the shifted Bragg wavelength) of the silicone embedded and bare FBG sensors are 0.949-0.959 and 0.869-0.939, respectively.
Manuscript v1_Figure 8. Linearity of Bragg wavelength when applying force on (a) silicone-embedded FBG sensors, and (b) bare FBG sensors.
Figure 8(2). Linearity of Bragg wavelength when applying force on (a) silicone-embedded FBG sensors, and (b) bare FBG sensors.
We understand that it may not be a good presentation of showing linearity, so we would like to change the x-axis from frequency (increment of 0.5N after every 5s) to force (N) to better highlight the differences in R2 between silicone-embedded and bare FBG sensors, and we have added linear trendlines for better comparison (Manuscript v2_Figure 7). Also, the corresponding equations (1-8) are shown before. (Line 264-280)
We have also included the second-order polynomials in Figure 7(2) to demonstrate the differences of using linear and polynomial linearity. We hope that these changes address your concerns.
Figure 7(2). Linearity of Bragg wavelength when applying force on (a) silicone-embedded FBG sensors, and (b) bare FBG sensors.
We appreciate your suggestion to use second-order polynomials in Manuscript v1_Figure 10, but after careful consideration, we have decided to continue using linear lines to present the R2. This will allow for easier comparison with the Plaince-X pressure sensor. Additionally, we have standardized the format of the graphs in Manuscript v2_Figure 9 by changing the x-axis and y-axis to shifted Bragg wavelength (nm) and force (N), respectively. The equations (1-4) have been updated to (9-12) based on the linear trendlines. (Line 338-350)
We hope that these changes meet with your approval.
Point 8: Nothing is said about how wearable this system is. You do not need just a sensor but also a laser light source and an optical system to measure the shift of the Bragg wavelength. I can hardly imagine a wearable interferometer. Or is this system only useful for people in a hospital who cannot move?
ANS 8: Thank you for your question regarding the wearable nature of the smart textile. We understand that using portable sensors is the main trend in this field, and we acknowledge the potential issues with the optical fiber's long tail and heavy interrogator in a clinical trial. Our next step is to develop a mini interrogator like FiSpec FBG X100 [1], which costs USD 200. We have added the Limitations of experiments and future works to the manuscript for greater clarity. Additionally, we have included the equipment setup in Manuscript v2_Figure 3 for better understanding. (Line 436-440)
We apologize for any confusion caused by the unclear background of the AIS and have provided a brief background in the Introduction. During the clinical trial, AIS patients will wear a tight-fit, pure cotton undergarment underneath the hard brace to reduce friction between the brace and the skin. They will be prescribed to stand, sit, and lie statically while adjusting the brace's tightness. Their mobility will be greatly restricted as they cannot bend their torso after donning the brace. This presents a valuable opportunity to use the smart undergarment during their fitting process. (Line 33-48)
Point 9: Extensive editing of English language required.
ANS 9: Thank you for taking the time to review my manuscript and provide valuable feedback. I appreciate your comments and suggestions, and I have carefully considered them in preparing my revised manuscript. Regarding the language quality, I understand that English is not my native language. I have already submitted my manuscript_v2 to a professional editing service (MDPI English editing) to address any errors. I am committed to improving the language quality of my manuscript and ensuring that it meets the standards of the journal. I hope that my revised manuscript reflects this commitment.
Reference
- FiSens. PRECISE, COMPACT AND COST-EFFECTIVE FiSpec FBG X100. Available online: https://fisens.com/products/fispec-fbg-x100 (accessed on 10 May).

Round 2
Reviewer 1 Report
I thank the authors for their detailed reply. I think now the paper is ready for publication.
Reviewer 2 Report
I am happy with the changes made by the authors. My conclusion is to accept this paper in its present form.